# Effectiveness of Maternal Inactivated COVID-19 Vaccination against Omicron Infection in Infants during the First 12 Months of Life: A Test-Negative Case-Control Study

**DOI:** 10.3390/vaccines11091402

**Published:** 2023-08-22

**Authors:** Jiayi Zhong, Wen Wang, Shuang Liu, Yifei Chen, Husheng Xiong, Xiang Meng, Dingmei Zhang, Yu Ma

**Affiliations:** 1Department of Epidemiology, School of Public Health, Sun Yat-sen University, Guangzhou 510080, China; 2Institute of Public Health, Guangzhou Medical University & Guangzhou Center for Disease Control and Prevention, Guangzhou 510440, China; 3NMPA Key Laboratory for Quality Monitoring and Evaluation of Vaccines and Biological Products, Guangzhou 510080, China

**Keywords:** maternal COVID-19 vaccination, infant infection, SARS-CoV-2, vaccine effectiveness, pregnancy

## Abstract

This study aims to evaluate the effectiveness of maternal inactivated COVID-19 vaccination before delivery for infants against Omicron infection in Guangzhou, China. A test-negative case-control design was conducted. This study selected infants born from 1 November 2021 to 23 November 2022 and tested for SARS-CoV-2 between 13 April 2022 and 30 November 2022 during outbreaks in Guangzhou. Multivariable logistic regression was performed to compare the maternal vaccination status of inactivated COVID-19 vaccines before delivery in cases and controls to estimate vaccine effectiveness (VE) for infants within 12 months. According to eligibility criteria, we finally selected 205 test-positive and 114 test-negative infants, as well as their mothers. The effectiveness of inactivated COVID-19 vaccines among fully vaccinated mothers was 48.4% (7.3% to 71.7%) for infants within 12 months, with the effectiveness of partial and booster vaccination showing no significant difference. Effectiveness for full vaccination presented a slight increase according to infants’ age at testing, with 49.6% (−12.3% to 78.4%) for 0–6 months and 59.9% (−0.6% to 84.4%) for over 6 months. A greater protective effect of two-dose vaccination was manifested in infants whose mother had received the second dose during the first trimester (65.9%, 95% CI: 7.7% to 87.9%) of pregnancy rather than preconception (43.5%, 95% CI: −8.7% to 71.1%). Moreover, VE could be improved to 77.1% (11.1% to 95.3%) when mothers received two doses both during pregnancy and 91.8% (41.1% to 99.6%) with receipt of a booster dose during pregnancy. Maternal vaccination with two doses of inactivated COVID-19 vaccines before delivery was moderately effective against Omicron infection in infants during the first 12 months of life. Full vaccination or a booster dose during pregnancy could confer better protection against Omicron for infants, although it might be overestimated due to the insufficient sample size in subgroups.

## 1. Introduction

Compared with adults, SARS-CoV-2 infection usually had mild or asymptomatic manifestations in children [1,2]. But in recent years, the emergence of the Omicron variant has been associated with increased hospitalizations among infants under 1 year old [3]. Up to now, COVID-19 vaccines have not been encouraged for infants in most countries; however, maternal antibodies via transplacental transfer were confirmed safe and effective for infants against severe SARS-CoV-2 infection, as well as hospitalizations [4,5,6,7].

Maternal antibodies induced by vaccination during pregnancy of mothers were well established for preventing viral infections in infants [8], such as influenza and respiratory syncytial virus (RSV) [9,10]. A similar process of maternal antibody transfer could occur with COVID-19 vaccination during pregnancy. SARS-CoV-2 antibodies were present in umbilical cord blood, breast milk, and serum specimens [11,12,13], which conferred protection for newborns with insufficient immunity after birth. Of note, maternal COVID-19 vaccination later in pregnancy were found to be associated with elevated IgG levels in infants after birth, with antibody levels persisting for approximately 6 months [14,15].

Previous studies about maternal antibodies focused mostly on mRNA COVID-19 vaccines in other countries [11,16,17]. So far, although China has offered COVID-19 vaccinations to young children aged 3–17, special groups like pregnant women (including women planning to conceive) and infants under the age of 3 are still not recommended to be vaccinated. Hence, whether the inactivated COVID-19 vaccines administered to pregnant women can confer effective protection against fetal infection has not yet been definitively concluded in China. Prior to the relaxation of control measures by the end of 2022, very few pregnant women and their infants were infected in China, so relevant data and studies are also lacking. 

In this test-negative case-control study, multivariable logistic regression was conducted to estimate the effectiveness of maternal partial vaccination, full vaccination, and booster vaccination before conception and during pregnancy in infants within the first 12 months of life. Through stratification, we further explored the optimal timing of maternal COVID-19 vaccination and observed the duration and attenuation of maternal protection in infants. These findings underline the importance of inactivated COVID-19 vaccination during pregnancy, and it is also of great practical significance to adjusting future COVID-19 vaccination strategies for mothers and infants in China and globally.

## 2. Materials and Methods

### 2.1. Study Design and Participants

We conducted this test-negative case-control study in Guangzhou, Guangdong Province, China, where newborns exceeded 360,000 in 2022. China initially began rolling out COVID-19 vaccination in late 2020. Hence, during the Guangzhou outbreaks in 2022, these infants born to vaccinated mothers would be approximately within 12 months (one year old) when they were tested for SARS-CoV-2. Therefore, we included 442 infants, born between 1 November 2021 and 23 November 2022, and their mothers. These infants were under unified management due to the Guangzhou outbreaks between 13 April 2022 and 30 November 2022. During this study period, Guangzhou had relatively strict control measures against COVID-19, which required all cases and close contacts to undergo quantitative real-time polymerase chain reaction (qPCR) test for SARS-CoV-2.

Infants who tested positive for SARS-CoV-2 were defined as the case group, and conversely, infants who tested negative and were identified as close contacts were defined as the control group. On this basis, we excluded multiple births, infants whose birth records and personal information were not available in the system database via their unique identification code (ID number), infants aged 12 months or older at testing, and infants whose mothers were positive for SARS-CoV-2 qPCR test postpartum. Our study was reviewed by the Guangzhou Center for Disease Control and Prevention and other participating institutions, which was determined to be public health surveillance and thus not subject to informed consent and ethical requirements.

This study focused on the protective effect on the fetus of pregnant women who were unvaccinated or vaccinated before delivery, regardless of whether they continued to vaccinate postpartum. Thus, we defined maternal partial vaccination as having received only one dose of inactivated COVID-19 vaccines at least 14 days prior to delivery (preconception or during pregnancy). Full vaccination of inactivated COVID-19 vaccines for mothers was designated as having received two doses at least 14 days prior to delivery, including two doses preconception or at least one dose during pregnancy. Maternal booster vaccination was defined as three doses received at least 14 days prior to delivery, which included three doses preconception or at least one dose during pregnancy. Correspondingly, mothers were considered unvaccinated if they had not received any doses before delivery, including mothers who were first vaccinated postpartum.

We, therefore, further excluded infants of mothers who received at least one dose of non-inactivated vaccines before delivery, such as COVID-19 vaccines based on viral vectors and recombinant cells, or sequential vaccination (Figure 1).

### 2.2. Data Sources and Collection

Firstly, the data we used were derived from Guangzhou Center for Disease Control and Prevention, which was responsible for identifying and tracking all COVID-19 cases and their close contacts during the outbreaks in Guangzhou, as well as collecting their demographic information, epidemiological history, and basic clinical information, such as case classification and symptoms of confirmed cases. COVID-19 cases referred to those who tested positive for SARS-CoV-2 via qPCR test. In contrast, close contacts were defined as persons who had a close contact with suspected cases and confirmed cases starting two days before the onset of symptoms or two days before the sampling of respiratory samples from an asymptomatic patient, without taking effective protection at the same time.

Moreover, we derived information on maternal COVID-19 vaccination from the database of the Guangdong Provincial Vaccine Circulation and Vaccination Management Information System. This was a centralized registration system about vaccines that contained the comprehensive vaccination records of all COVID-19 vaccines for the permanent population in Guangdong Province. We identified each mother–newborn pair via the participants’ ID numbers, and then recorded the date of each dose of COVID-19 vaccination and the manufacturer. Almost 81% of the infants’ and mothers’ information was successfully found and read, while 19% of the records contained missing data.

In our study, none of the infants included were found to have received any COVID-19 vaccines. Moreover, as the pregnancy progressed, few pregnant women in China chose to continue vaccinating in the second and third trimesters of pregnancy. Therefore, we did not find mothers who received any dose of COVID-19 vaccines up to 14 days before delivery. During the study period, the interval between the first dose and second dose of maternal full vaccination of inactivated COVID-19 vaccines ranged from 3 to 8 weeks (21–56 days). Additionally, some pregnant women had also received heterologous doses of inactivated COVID-19 vaccines due to fluctuations in vaccine supply. In China, the booster dose was introduced in December 2021 and gradually extended to all adults, but it was also not applicable to pregnant women and infants.

### 2.3. Outcomes and Covariates

Laboratory-confirmed SARS-CoV-2 infection in infants was defined as a positive result of a qPCR test from the collected respiratory specimen, regardless of severity or the presence of related clinical symptoms. Furthermore, GZCDC sequenced all participant samples with positive qPCR results of SARS-CoV-2, thus confirming that the SARS-CoV-2 variant circulating in Guangzhou during the study period was Omicron.

We selected the covariates to be adjusted based on their potential associations with infection outcomes in infants and maternal inactivated COVID-19 vaccination status. For infants and their mothers, the following covariates were extracted from the electronically recorded data from Guangzhou Center for Disease Control and Prevention and the Guangdong Provincial Vaccine Circulation and Vaccination Management Information System: infant age at testing (months) (continuous), infant sex, and mother’s age at birth (years) (continuous), etc. All adjusted variables were selected a priori based on previous work [18].

### 2.4. Power Analysis and Sample Size Calculation

The sample size was decided on the basis of the formula applicable to the case-control study, n = 2p
q (Z_1-α/2_ + Z_β_)^2^/(p_1_ − p_0_)^2^. The threshold of α was set at 0.05 and that of β was set at 0.10. The odds ratio (OR) for inactivated COVID-19 vaccination was estimated to be 0.40 (VE = 60%), and the vaccination rate of the control group (p_0_) was 60%. Thus, we calculated that the minimum sample size was 212 (106 per group), which provided 90% statistical power. 

Likewise, via power analysis, all *p*-values in statistical tests were provided with their effect sizes (ES) according to the specified power (90%) and the sample size. For different statistical tests, the selected measures of effect sizes included Cohen’s d or Hedges’ g, f, η^2^, and ω. Referring to Cohen’s standard, we defined the ES as small effect, intermediate effect, and strong effect with 0.1, 0.3, and 0.5 as boundaries, respectively.

### 2.5. Statistical Analysis

We used percentages for categorical data, medians (interquartile ranges), or means (standard deviations) for continuous data to describe the characteristics of study participants. Chi-squared test and Fisher’s exact test were used for categorical data, while *t*-test, analysis of variance (ANOVA), and Kruskal–Wallis H test (including post-hoc pairwise comparisons based on the Bonferroni) were used for continuous data. *p*-values were used to assess differences between case and control groups and between maternally vaccinated and unvaccinated groups.

Meanwhile, multivariable logistic regression model was established to estimate the adjusted ORs of maternal COVID-19 vaccination status between cases and controls. We adjusted the covariates mentioned in the previous section to calculate vaccine effectiveness as (1-adjusted OR) × 100%. The study primarily calculated vaccine effectiveness for maternal full vaccination and modelled infant age at testing (months) using restricted cubic splines with four knots (Figure 2). To assess the potential decline in effectiveness over time after full vaccination, we stratified the participants by infant age at testing (i.e., 0–6 months and >6 months) and the timing of the second dose received (i.e., preconception, first trimester and second trimester of pregnancy). Additional analyses were conducted for partial vaccination, booster vaccination, and the timing of doses of fully and booster vaccination received before delivery using infants who tested negative as the control groups.

Four sensitivity analyses for maternal full vaccination against Omicron infection were conducted: including infants of mothers who tested positive for SARS-CoV-2 postpartum, as mothers might transfer antibodies to infants through breastfeeding postpartum; excluding infants of mothers who were first vaccinated postpartum (i.e., unvaccinated before delivery); excluding infants hospitalized within the first 12 months of life; excluding infants of mothers who tested positive for Hepatitis B surface antigen (HBsAg), since these mothers might be unable to mount an adequate immune response after two-dose vaccination due to underlying diseases.

We used R software (version 4.2.3) for all analysis. Tests were two-sided, and *p* < 0.05 indicated a statistically significant imbalance between groups. A confidence interval of 95% (95% CI) was used to show the precision of the OR.

## 3. Results

### 3.1. Study Population

Of 442 infants who were under unified management due to the outbreaks between 13 April 2022 and 30 November 2022 in Guangzhou, 319 finally met the eligibility criteria. During the first 12 months of life, 205 infants tested positive for SARS-CoV-2 by qPCR (the case group) and 114 infants tested negative (the control group) (Figure 1). More than half (50.8%, 162 of 319) of the study participants were aged 0-6 months when they were tested for SARS-CoV-2. Mothers of younger infants at testing were more likely to be vaccinated with multiple doses before delivery. Moreover, 112 test-positive infants (54.6%) were cases with mild symptoms, of which fever (41.0%) was the most common, and 14 infants (6.8%) were hospitalized for reasons related to COVID-19.

A total of 65.5% of mothers were aged between 25 and 34 years at birth, and the mothers of infants in the case group were younger than that in the control group (31.20 years vs. 32.86 years, *p* = 0.004). Nevertheless, between different vaccination status groups, there were no statistical differences in the mothers’ age at birth (*p* > 0.05). More infants (33.9%, 108 of 319) were born to mothers who had been fully vaccinated before delivery, with 5.6% (18 of 319) partially vaccinated and 4.1% (13 of 319) having received a booster vaccination. Of the 319 infants included, 56.4% of their mothers were unvaccinated prior to delivery. Additionally, more vaccinated mothers received inactivated COVID-19 vaccines before conception (71.9%, 100 of 139) than during pregnancy (28.1%, 39 of 139). The majority (72.2%, 13 of 18) of partial vaccinations was administered during the first trimester of pregnancy. Baseline characteristics of infants and mothers were shown in Table 1.

Notably, in the case and control groups, the effect sizes for maternal vaccination variables were 0.30 and above, of which the third dose showed a strong effect (ES = 0.90).

### 3.2. Vaccine Effectiveness for Maternal Full Vaccination

After adjusting for covariates, maternal full vaccination of inactivated COVID-19 vaccines before delivery significantly reduced the risk of Omicron infection by 48.4% (95% CI: 7.3% to 71.7%) in infants during the first 12 months of life (Table 2). Nonetheless, the effectiveness of full vaccination exhibited a slight increase with the advancing age of infants at testing, as demonstrated by effectiveness ranging from 49.6% (−12.3% to 78.4%) for infants aged 0–6 months to 59.9% (−0.6% to 84.4%) for those older than 6 months. Meanwhile, the efficacy of the second dose received within 12 weeks of gestation (i.e., during the first trimester of pregnancy, 65.9% (7.7% to 87.9%)) was greater than that received preconception (43.5% (−8.7% to 71.1%)) (Figure 3).

### 3.3. Additional Analysis and Sensitivity Analysis

We performed repeated analyses of vaccine effectiveness for partial vaccination and booster vaccination, despite the fact that both results were not statistically significant. During the Omicron period, among infants whose mothers received two doses, both during pregnancy, VE against infection was 77.1% (95% CI: 11.1% to 95.3%) during the first 12 months of life (Table 3). For children whose mothers received two doses before pregnancy and one booster dose during pregnancy, VE against infection could reach 91.8% (95% CI: 41.1% to 99.6%) up to 12 months of life. These results suggest that both full and booster vaccination during pregnancy could provide more superior protection against Omicron for infants, although this finding might be overestimated due to the insufficient sample size in subgroups.

Sensitivity analyses showed that the models we constructed were relatively robust, with vaccine effectiveness for full vaccination against Omicron infection ranging from 43.9% to 65.0% (Table 4).

## 4. Discussion

Pregnant women were at higher risk of COVID-19-related complications than their non-pregnant counterparts [19,20]. To date, no significant adverse outcomes have been found in pregnant women after COVID-19 vaccination [21]. Conversely, epidemiological studies have suggested that COVID-19 vaccination of pregnant women was very effective in preventing severe SARS-CoV-2 infection and adverse pregnancy outcomes, such as spontaneous abortion, stillbirth, and premature delivery [22,23,24]. Previously, many studies have indicated that COVID-19 vaccination during pregnancy might have a dual benefit, as it could also transfer antibodies against SARS-CoV-2 to the fetus via the placenta or breast milk and provide immune protection to the neonates after birth [13,14,25,26].

In our study, we used test-negative case-control study design to estimate that the maternal partial vaccination of inactivated COVID-19 vaccines before conception and during pregnancy was 48.4% effective in preventing Omicron infection in infants during the first 12 months of life. Our study also indicated that the protection provided by fully vaccinated mothers for their babies did not diminish over time after infants’ birth within 12 months. Moreover, the best protective effect of full vaccination was manifested in infants whose mother had received the second dose during pregnancy rather than preconception, especially two doses both administered during pregnancy. Similarly, with receipt of only a booster dose during pregnancy, vaccine effectiveness could be improved to 91.8%, although this is likely to be overestimated due to the limited sample size in this subgroup. The clinical implications of these findings for maternal vaccination need to be weighed against the risks of being vaccinated later during pregnancy to mothers and the fetus.

Our results were broadly consistent with two recent studies. These studies reported that mRNA COVID-19 vaccination of mothers during pregnancy was effective in reducing Delta or Omicron infection [4,27]. In a case-control study that included 8809 newborns, mothers who received two and three doses of mRNA COVID-19 vaccines during pregnancy had a 45% and 73% lower risk of infection, respectively, during the Omicron-dominant period [4]. Additionally, we found that infants whose mothers received the booster dose during pregnancy were related to a lower risk of testing positive for Omicron than those who received two doses during pregnancy, similar to a Norwegian cohort study [28]. During the Delta or Omicron-predominant period, estimates of VE were found to be higher when mothers were vaccinated during pregnancy, especially the second or third trimester [29]. Slightly different from that, we emphasized the first trimester of pregnancy compared to preconception, since few pregnant women in China had received COVID-19 vaccines in the second or third trimester. We could not accurately estimate vaccine effectiveness in these two periods.

Contrary to the conclusions of most studies that vaccine effectiveness declined over time after infants’ birth [4,27,30], this study found that the effectiveness of maternal full vaccination was higher in infants over 6 months after birth than that in younger infants, which seemed out of the ordinary. However, this conclusion had a relatively large confidence interval and was not statistically significant and might be related to the insufficient sample size of infants in our study.

Our research, absolutely, had limitations inevitably. We adjusted for several potential confounding factors, but it was limited by the information available in the public surveillance database, and some potential confounders (e.g., maternal delivery and breastfeeding information) might also differ between infants of vaccinated and unvaccinated mothers. Moreover, the mothers who were first vaccinated postpartum were also likely to pass SARS-CoV-2 antibodies to their fetus and provide protection through breastfeeding after delivery. Nevertheless, since the database did not have specific records on breastfeeding, we cannot delve into how COVID-19 infection in infants was affected by breastfeeding. For this reason, this study emphasized the protection for infants from maternal antibodies transferred through the placenta after prenatal vaccination, rather than via breast milk. Furthermore, due to COVID-19 vaccination strategies in China, we were unable to estimate the protective effect of inactivated COVID-19 vaccines during the second and third trimesters of pregnancy. Our study was also limited to assessing inactivated vaccines, as few pregnant women in China received COVID-19 vaccines based on viral vectors or other non-inactivated types during the study period.

However, our research strengths mainly lay in the reliability of the data sources and the innovation of the topic. During the study period, infants infected with SARS-CoV-2 or identified as close contacts in outbreaks received a unified qPCR test and isolation control measures and were accurately registered in GZCDC’s public surveillance database, which ruled out the effect of rapid antigen test results at home on missing information. A significant proportion of mothers in our study had already been vaccinated before or during pregnancy. Through the population-based database in Guangdong Provincial Vaccine Circulation and Vaccination Management Information System, we ensured highly accurate classification of each mother’s vaccination status from the detailed information the system provided, which was linked using the unique ID number. At the same time, we used a test-negative case-control design to effectively reduce various bias caused by different test methods, healthcare-seeking behaviors, and case diagnosis.

Since 1 December 2022, Guangzhou, as well as other cities in China, have gradually begun to optimize and relax previous COVID-19 control measures. Before that, the number of COVID-19 cases in China had been maintaining at a low level due to relatively strict control measures, with even fewer amounts of infant cases. Meanwhile, relevant data on pregnant women who received inactivated COVID-19 vaccines during pregnancy in China are also limited. Hence, this study is innovative and of great practical significance, which can provide more references for future clinical recommendations and maternal–infant vaccination strategies of inactivated COVID-19 vaccines in China.

## 5. Conclusions

In this test-negative case-control study, maternal vaccination with two doses of inactivated COVID-19 vaccines before delivery was moderately effective (48.4%) against Omicron infection in infants within 12 months. Maternal full vaccination during the first and second trimester of pregnancy could confer more excellent protection for infants than that received before conception. With receipt of a booster dose during pregnancy, VE could be improved to 91.8%. Moreover, the effectiveness of maternal full vaccination was higher in infants aged over 6 months compared to those aged 0-6 months, which might be related to the inadequate sample size.

## Figures and Tables

**Figure 1 vaccines-11-01402-f001:**
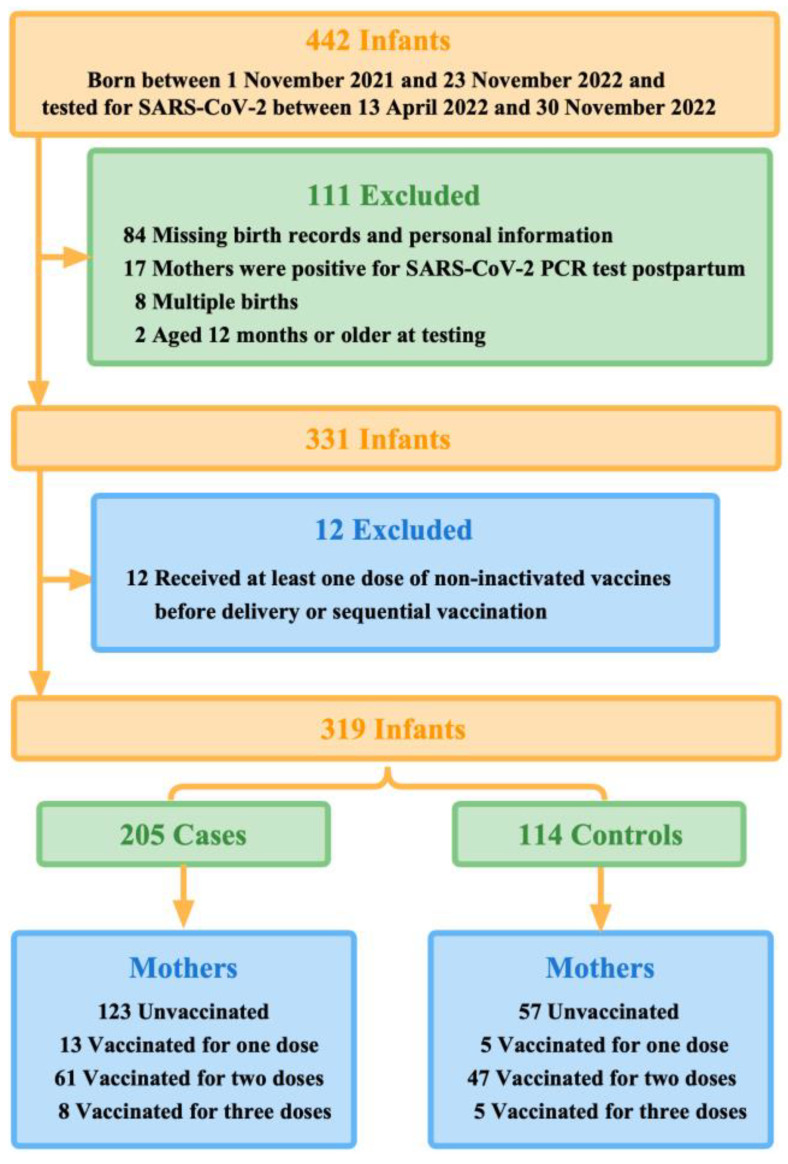
Summary of study participants selection.

**Figure 2 vaccines-11-01402-f002:**
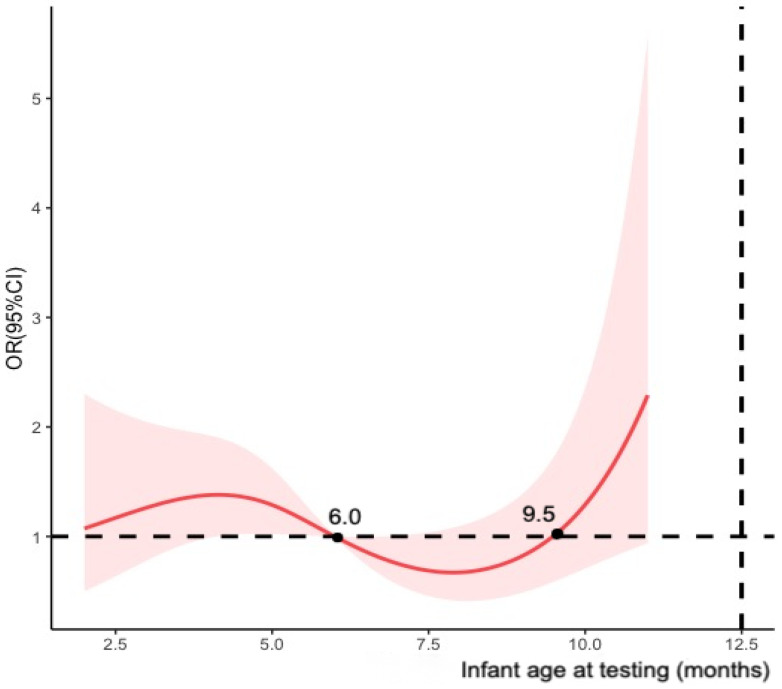
Restricted cubic splines modelled on infant age at testing (four knots).

**Figure 3 vaccines-11-01402-f003:**
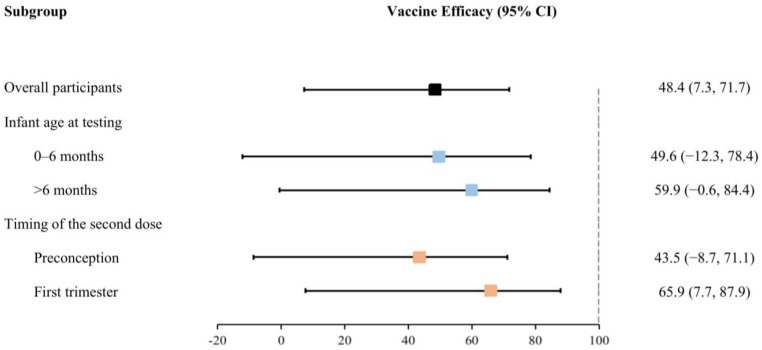
Effectiveness of the full vaccination of inactivated COVID-19 vaccines for infants against Omicron infection (stratified by infant age at testing and the timing of the second dose received).

**Table 1 vaccines-11-01402-t001:** Characteristics of infants during the first 12 months of life, tested for SARS-CoV-2 between 13 April 2022 and 30 November 2022, and their mothers, Guangzhou, China.

Characteristic	Infants Tested for SARS-CoV-2			Maternal COVID-19 Vaccination Status				
	Case (n = 205) (%)	Control (n = 114) (%)	*p*-Value (Effect Size)	Unvaccinated ^1^ (n = 180) (%)	Partially ^2^ (n = 18) (%)	Fully ^3^ (n = 108) (%)	Booster ^4^ (n = 13) (%)	*p*-Value (Effect Size)
Infant sex, female	91 (44.4)	59 (51.8)	0.252 (0.18)	93 (51.7)	4 (22.2)	48 (44.4)	5 (38.5)	0.083 (0.23)
Infant age at testing (months), median (interquartile range)	6.00 (4.00, 10.00)	7.00 (4.00, 9.00)	0.764 (0.38)	9.00 (5.75, 10.00)	7.50 (5.25, 9.00)	4.00 (3.00, 6.00)	1.00 (1.00, 2.00)	<0.001 ^a^ (0.27)
Infant’s age category (months)			0.428 (0.18)					<0.001 (0.21)
0–6	108 (52.7)	54 (47.4)		57 (31.7)	7 (38.9)	85 (78.7)	13 (1.0)	
>6	97 (47.3)	60 (52.6)		123 (68.3)	11 (61.1)	23 (21.3)	0 (0.0)	
Symptoms of confirmed cases			-					0.825 ^b^ (0.37)
Fever	84 (41.0)	-		41 (22.8)	8 (44.4)	33 (30.6)	2 (15.4)	
Cough	22 (10.7)	-		13 (7.2)	1 (5.6)	8 (7.4)	0 (0.0)	
Classification of cases			-					0.007 ^b^ (0.26)
Mild	112 (54.6)	-		56 (31.1)	9(50.0)	43 (39.8)	4 (30.8)	
Asymptomatic	93 (45.4)	-		67 (37.2)	4 (22.2)	18 (16.7)	4 (30.8)	
Admission to hospital related to COVID-19	14 (6.8)	-	-	3 (1.7)	3 (16.7)	7 (6.5)	1 (7.7)	-
Mother’s age at birth (years), mean (standard deviations)	31.20 (5.04)	32.86 (4.83)	0.004 (0.33)	31.54 (5.16)	30.72 (4.89)	32.19 (4.68)	33.54 (5.90)	0.320 (0.21)
Mother’s age category (years)			0.017 (0.20)					0.283 ^b^ (0.23)
≤24	13 (6.3)	4 (3.5)		14 (7.8)	1 (5.6)	2 (1.9)	0 (0.0)	
25–34	143 (69.8)	66 (57.9)		115 (63.9)	13 (72.2)	74 (68.5)	7 (53.8)	
35–46	49 (23.9)	44 (38.6)		51 (28.3)	4 (22.2)	32 (29.6)	6 (46.2)	
Timing of full vaccination received before delivery			0.247 ^b^ (0.34)					-
Two doses all received before pregnancy	52 (25.4)	36 (31.6)		0 (0.0)	0 (0.0)	88 (81.5)	0 (0.0)	
One dose before pregnancy and one dose during pregnancy	6 (2.9)	4 (3.5)		0 (0.0)	0 (0.0)	10 (9.3)	0 (0.0)	
Two doses all received during pregnancy	3 (1.5)	7 (6.1)		0 (0.0)	0 (0.0)	10 (9.3)	0 (0.0)	
Manufacturers of maternal vaccination dose 1 and 2			0.568 (0.32)					-
CNBG-CNBG ^5^	18 (8.8)	10 (8.8)		0 (0.0)	0 (0.0)	26 (24.1)	2 (15.4)	
Sinovac R&D-Sinovac R&D ^5^	31 (15.1)	23 (20.2)		0 (0.0)	0 (0.0)	47 (43.5)	7 (53.8)	
Heterologous ^6^	20 (9.8)	19 (16.7)		0 (0.0)	0 (0.0)	35 (32.4)	4 (30.8)	
Timing of maternal vaccination								
First dose			0.237 ^b^ (0.30)					-
Preconception	71 (34.6)	45 (39.5)		0 (0.0)	5 (27.8)	98 (90.7)	13 (100.0)	
First trimester	10 (4.9)	12 (10.5)		0 (0.0)	13 (72.2)	9 (8.3)	0 (0.0)	
Second trimester	1 (0.5)	0 (0.0)		0 (0.0)	0 (0.0)	1 (0.9)	0 (0.0)	
Second dose			0.207 ^b^ (0.32)					-
Preconception	60 (29.3)	41 (36.0)		0 (0.0)	0 (0.0)	88 (81.5)	13 (100.0)	
First trimester	8 (3.9)	11 (9.6)		0 (0.0)	0 (0.0)	19 (17.6)	0 (0.0)	
Second trimester	1 (0.5)	0 (0.0)		0 (0.0)	0 (0.0)	1 (0.9)	0 (0.0)	
Third dose			0.005 ^b^ (0.90)					-
Preconception	7 (3.4)	0 (0.0)		0 (0.0)	0 (0.0)	0 (0.0)	7 (53.8)	
First trimester	1 (0.5)	5 (4.4)		0 (0.0)	0 (0.0)	0 (0.0)	6 (46.2)	
Interval between doses 1 and 2			0.261 ^b^ (0.32)					-
21–34 days	51 (24.9)	35 (30.7)		0 (0.0)	0 (0.0)	76 (70.4)	10 (76.9)	
35–55 days	17 (8.3)	13 (11.4)		0 (0.0)	0 (0.0)	27 (25.0)	3 (23.1)	
≥56 days	1 (0.5)	4 (3.5)		0 (0.0)	0 (0.0)	5 (4.6)	0 (0.0)	
Interval between doses 2 and 3			0.103 ^b^ (0.90)					-
180–193 days	6 (2.9)	1 (0.9)		0 (0.0)	0 (0.0)	0 (0.0)	7 (53.8)	
≥194 days	2 (1.0)	4 (3.5)		0 (0.0)	0 (0.0)	0 (0.0)	6 (46.2)	

Note: *p*-values comparing different groups were from t-test, chi-squared test, and analysis of variance (ANOVA). ^a^ Kruskal–Wallis H test. Results of the Bonferroni in post hoc tests indicated that all differences of infants’ age at testing between any two groups in maternal vaccination status were significant (adjusted *p* < 0.05), except for the unvaccinated group and the partial vaccinated group. ^b^ Fisher’s exact test. Significance difference: *p* < 0.05. The “-” indicates no data. ^1^ Unvaccinated mothers: having not received any doses before delivery (preconception or during pregnancy), including mothers who were first vaccinated postpartum. ^2^ Maternal partial vaccination: having received only one dose of inactivated COVID-19 vaccines at least 14 days prior to delivery. ^3^ Maternal full vaccination: having received two doses at least 14 days prior to delivery, including two doses preconception or at least one dose during pregnancy. ^4^ Maternal booster vaccination: having received three doses received at least 14 days prior to delivery, including three doses preconception or at least one dose during pregnancy. ^5^ CNBG: China National Biotec Group. Sinovac R&D: Sinovac Research & Development Co., Ltd. ^6^ Heterologous: CNBG followed by Sinovac R&D or vice versa.

**Table 2 vaccines-11-01402-t002:** Effectiveness of maternal full vaccination received at different timing for infants during the first 12 months of life against Omicron infection in Guangzhou, China.

	Infants in Case Group (n = 205) (%)	Infants in Control Group (n = 114) (%)	*p*-Value	Adjusted Odds Ratio (OR) ^1^ and 95% Confidence Interval (95% CI)	Adjusted Effectiveness of Full Vaccination ^1^ (95% CI)
Overall participants			0.028		
Unvaccinated ^2^	123 (60.0)	57 (50.0)		Reference	Reference
Received full vaccination ^3^	61 (29.8)	47 (41.2)		0.516 (0.283, 0.927)	48.4 (7.3, 71.7)
Infant age at testing					
0–6 months			0.101		
Unvaccinated ^2^	43 (21.0)	14 (12.3)		Reference	Reference
Received full vaccination ^3^	51 (24.9)	34 (29.8)		0.504 (0.216, 1.123)	49.6 (−12.3, 78.4)
>6 months			0.053		
Unvaccinated ^2^	80 (39.0)	43 (37.7)		Reference	Reference
Received full vaccination ^3^	10 (4.9)	13 (11.4)		0.401 (0.156, 1.006)	59.9 (−0.6, 84.4)
Timing of the second dose					
Preconception			0.090		
Unvaccinated ^2^	123 (60.0)	57 (50.0)		Reference	Reference
Received full vaccination ^3^	52 (25.4)	36 (31.6)		0.565 (0.289, 1.087)	43.5 (−8.7, 71.1)
First trimester			0.035		
Unvaccinated ^2^	123 (60.0)	57 (50.0)		Reference	Reference
Received full vaccination ^3^	8 (3.9)	11 (9.6)		0.341 (0.121, 0.923)	65.9 (7.7, 87.9)
Second trimester			-		
Unvaccinated ^2^	123 (60.0)	57 (50.0)		Reference	Reference
Received full vaccination ^3^	1 (0.5)	0 (0.0)		-	-

^1^ ORs and VE were adjusted for infant age at testing (months) (continuous), infant sex, and mother’s age at birth (years) (continuous). The “-” indicates no data or cannot be analyzed due to the insufficient sample size in this subgroup. Significant difference: *p* < 0.05. ^2^ Unvaccinated mothers: having not received any doses before delivery (preconception or during pregnancy), including mothers who were first vaccinated postpartum. ^3^ Maternal full vaccination: having received two doses at least 14 days prior to delivery, including two doses preconception or at least one dose during pregnancy.

**Table 3 vaccines-11-01402-t003:** Additional analyses of effectiveness for maternal inactivated COVID-19 vaccination in Guangzhou, China.

	Infants in Case Group (n = 205) (%)	Infants in Control Group (n = 114) (%)	*p*-Value	Adjusted Odds Ratio (OR) ^1^ and 95% Confidence Interval (95% CI)	Adjusted Effectiveness of Maternal Vaccination ^1^ (95% CI)
Overall participants					
Unvaccinated ^2^	123 (60.0)	57 (50.0)		Reference	Reference
Received partial vaccination ^3^	13 (6.3)	5 (4.4)	0.955	0.969 (0.329, 3.247)	3.1 (−224.7, 67.1)
Received booster vaccination ^4^	8 (3.9)	5 (4.4)	0.683	0.753 (0.195, 3.089)	24.7 (−208.9, 80.5)
Timing of full vaccination ^5^ received before delivery					
Unvaccinated ^2^	123 (60.0)	57 (50.0)		Reference	Reference
Two doses all received before pregnancy	52 (25.4)	36 (31.6)	0.090	0.565 (0.289, 1.087)	43.5 (−8.7, 71.1)
One dose before pregnancy and one dose during pregnancy	6 (2.9)	4 (3.5)	0.644	0.726 (0.190, 3.070)	27.4 (−207.0, 81.0)
Two doses all received during pregnancy	3 (1.5)	7 (6.1)	0.042	0.229 (0.047, 0.889)	77.1 (11.1, 95.3)
Timing of booster vaccination ^4^ received before delivery					
Unvaccinated ^2^	123 (60.0)	57 (50.0)		Reference	Reference
Three doses all received before pregnancy	7 (3.4)	0 (0.0)	-	-	-
Two doses before pregnancy and only booster dose during pregnancy	1 (0.5)	5 (4.4)	0.031	0.082 (0.004, 0.589)	91.8 (41.1, 99.6)

^1^ ORs and VE were adjusted for infant age at testing (months) (continuous), infant sex, and mother’s age at birth (years) (continuous). The “-” indicates no data or cannot be analyzed due to the insufficient sample size in this subgroup. Significance difference: *p* < 0.05. ^2^ Unvaccinated mothers: having not received any doses before delivery (preconception or during pregnancy), including mothers who were first vaccinated postpartum. ^3^ Maternal partial vaccination: having received only one dose of inactivated COVID-19 vaccines at least 14 days prior to delivery. ^4^ Maternal booster vaccination: having received three doses received at least 14 days prior to delivery, including three doses preconception or at least one dose during pregnancy. ^5^ Maternal full vaccination: having received two doses at least 14 days prior to delivery, including two doses preconception or at least one dose during pregnancy.

**Table 4 vaccines-11-01402-t004:** Sensitivity analyses of effectiveness for maternal full vaccination against Omicron infection in infants during the first 12 months of life, Guangzhou, China.

	Case Infants	Control Infants	Adjusted OR ^1^	Effectiveness of Full Vaccination ^1^
	No. Vaccinated Mothers/Total No. (%)		(95% Confidence Interval)	% (95% Confidence Interval)
Model 1 ^2^	70/222 (31.5)	47/114 (41.2)	0.561 (0.311, 1.000)	43.9 (0.0, 68.9)
Model 2 ^3^	61/188 (32.4)	47/87 (54.0)	0.350 (0.181, 0.660)	65.0 (34.0, 81.9)
Model 3 ^4^	54/191 (28.3)	47/114 (41.2)	0.475 (0.259, 0.858)	52.5 (14.2, 74.1)
Model 4 ^5^	54/191 (28.3)	42/102 (41.2)	0.481 (0.253, 0.897)	51.9 (10.3, 74.7)

^1^ Maternal full vaccination referred to having received two doses at least 14 days prior to delivery, including two doses preconception or at least one dose during pregnancy. ORs and VE were adjusted for infant age at testing (months) (continuous), infant sex, and mother’s age at birth (years) (continuous). Significance difference: *p* < 0.05. ^2^ Infants of mothers who tested positive for SARS-CoV-2 postpartum included. ^3^ Infants of mothers who were first vaccinated postpartum (i.e., unvaccinated before delivery) excluded. ^4^ Infants hospitalized within the first 12 months of life excluded. ^5^ Infants of mothers who tested positive for HBsAg excluded.

## Data Availability

Not applicable.

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
