# Peer review of "Effectiveness of Maternal Inactivated COVID-19 Vaccination against Omicron Infection in Infants during the First 12 Months of Life: A Test-Negative Case-Control Study"

_vaccines, 2023, doi:10.3390/vaccines11091402_

Round 1

Reviewer 1 Report

Well planned work.

A few points need to be corrected.

The sample size was decided on the basis of which criteria.

Which of the post hoc tests were used for ANOVA and Kruskal Wallis?

Author Response

Point 1: Well planned work. A few points need to be corrected. The sample size was decided on the basis of which criteria.

Response 1: Thank you for pointing out this problem. We have supplemented the procedure and results of sample size calculation in the Materials and Methods section, as well as the criterion based on a case-control study design (page 4, lines 153-157). The calculation results showed that the sample size of this study met the requirements (n=212, 106 per group).

Point 2: Which of the post hoc tests were used for ANOVA and Kruskal Wallis?

Response 2: We deeply appreciate your suggestion. We have added relevant statements and the results of the post-hoc pairwise comparisons based on the Bonferroni for Kruskal-Wallis H test (page 4, lines 167-168 and page 7, lines 246-249). Since the post-hoc test should be used when there is a statistical difference between multiple groups (P<0.05), the ANOVA test adopted in this study did not need to complete this step (P=0.320). Regarding this point, we also made a supplementary explanation in the manuscript (page 5, lines 232-233). Thank you for pointing out this problem.

Reviewer 2 Report

The manuscript is relevant and well-written. Before it gets published, I suggest to re-write some of the methodology, it contains part of introductory elements (lines 126-130) and results (130-136).

Author Response

Point 1: The manuscript is relevant and well-written. Before it gets published, I suggest to re-write some of the methodology, it contains part of introductory elements (lines 126-130) and results (130-136).

Response 1: Thanks a lot for this constructive suggestion. Based on your suggestions, We have rewritten the relevant sentences (page 4, lines 128-132) and placed some of them in the more appropriate Introduction section (page 2, lines 57-60) and the Results section (page 6, lines 236-238).

Reviewer 3 Report

This is a very interesting study aimed to evaluate the effectiveness of maternal inactivated COVID-19 vaccination before delivery for infants against Omicron infection in Guangzhou, China. A test-negative case-control design was performed. According to eligibility criteria, they finally selected 205 test-positive and 114 test-negative infants, as well as their mothers and compaired them. The authors gave detailed description of the obtained results. However, some major revisions are needed.

Comment 1: In the Material and Methods section, the “Power analysis and sample size calculation” paragraph should be added, before statistical analysis paragraph. In relation to the other statistical weaknesses, I suggest that the effect size should be given for all statistical tests. Explorative analyses are legitimate but need to be labelled as such p-values can only be interpreted by alone in confirmatory analyses; in exploratory analyses, p-values should be completed with effect sizes.

Comment 2: Line 101-102: The authors state that „Correspondingly, mothers were considered unvaccinated if they had not received any doses before delivery, including mothers who were first vaccinated postpartum.“ In my opinion, it would have been better to include mothers who were vaccinated only after delivery in the vaccinated group. It is well known that mothers might transfer antibodies to infants through breastfeeding postpartum, which could modify the results described. Please justify your decision, or I suggest re-analysing the data to include these mothers in the vaccinated group.

Comment 3: My comment is related to the previous comment. Do you have any information about bresfeeding of these mothers and their infants? It would be interesting to examine how COVID-19 infection in infants is affected by breastfeeding, i.e. how breastfeeding affects the development of infection.

Comment 4: In the case of Table 2, the "p" values are missing. Please calculate and write in these values, as this will give us an understanding of whether there is a significant difference or relationship between the parameters in the table.

In the same table, the "Adjusted effectiveness of fully vaccination", why "×100%" is given in brackets. Is this not just a simple percentage?

Author Response

Point 1: The authors gave detailed description of the obtained results. However, some major revisions are needed. In the Material and Methods section, the “Power analysis and sample size calculation” paragraph should be added, before statistical analysis paragraph. In relation to the other statistical weaknesses, I suggest that the effect size should be given for all statistical tests. Explorative analyses are legitimate but need to be labelled as such p-values can only be interpreted by alone in confirmatory analyses; in exploratory analyses, p-values should be completed with effect sizes.

Response 1: Thank you for your valuable suggestion. We have added “Power analysis and sample size calculation” paragraph and performed power analysis for the sample size and effect sizes for all statistical tests (page 4, lines 152-162). In addition, the results on the effect sizes for all P-values were presented in Table 1 and page 6, lines 241 and 242. The effect sizes of Table 2-Table 4 were OR, which have been analyzed and listed in the tables.

Point 2: Line 101-102: The authors state that “Correspondingly, mothers were considered unvaccinated if they had not received any doses before delivery, including mothers who were first vaccinated postpartum.“ In my opinion, it would have been better to include mothers who were vaccinated only after delivery in the vaccinated group. It is well known that mothers might transfer antibodies to infants through breastfeeding postpartum, which could modify the results described. Please justify your decision, or I suggest re-analysing the data to include these mothers in the vaccinated group.

Response 2: Thank you for these comments. Previous studies have indeed found that antibodies produced by COVID-19 vaccination could be detected in breast milk, which might provide protection for infants. However, the main component in breast milk is IgA, while the IgG passed through the placenta is the main component of maternal antibodies. Therefore, the purpose of our study was to examine the protective effect on the fetus of pregnant women who were unvaccinated or vaccinated before delivery, regardless of whether they continued to vaccinate postpartum (page 1, lines 17-18 and page 2, lines 94-96). We emphasized the influence of antibodies passed through the placenta after prenatal vaccination, rather than breast milk.

Equally important, due to the lack of follow-up information and exact data on breastfeeding, we cannot determine clearly whether and for how long each mother was breastfeeding (including the average amount of breast milk). Therefore, we think it is not appropriate to classify these mothers as vaccinated group directly. Meanwhile, we have supplemented this concern as a limitation in the Discussion section (page 11, lines 370-375).

Point 3: My comment is related to the previous comment. Do you have any information about breastfeeding of these mothers and their infants? It would be interesting to examine how COVID-19 infection in infants is affected by breastfeeding, i.e. how breastfeeding affects the development of infection.

Response 3: Thank you for this great suggestion. It’s really a good idea to examine how the protection was affected by breastfeeding. Regrettably, in this study, we did not collect detailed information on mothers' breastfeeding postpartum, as public surveillance database did not have relevant records. So, an in-depth analysis of this scientific question is relatively difficult in practice. Absolutely, it is also related to our research purpose and scope. The purpose of our study focused on the effectiveness of maternal vaccination prior to delivery, but not breastfeeding postpartum. We have added these statements combined with the previous comment in the Discussion section (page 11, lines 370-375). Thanks a lot for your kind suggestion.

Point 4: In the case of Table 2, the "p" values are missing. Please calculate and write in these values, as this will give us an understanding of whether there is a significant difference or relationship between the parameters in the table. In the same table, the "Adjusted effectiveness of fully vaccination", why "×100%" is given in brackets. Is this not just a simple percentage?

Response 4: Thank you for your valuable suggestion. We have calculated and added the "P-values" for ORs in Table 2 and Table 3. In the same table, the original meaning of “×100%” in brackets was to show that VE=(1-adjusted OR)×100%. We have revised this statement to a more appropriate “95% CI“ (Table 2 and Table 3).

Round 2

Reviewer 3 Report

Thank you very much for your comprehensive, detailed and accurate answers. With the changes you have made, you have created a really interesting, valuable, easy to understand paper with new information that will be useful for other researchers in the future. Congratulations.